# Physical Activity Advice and Counselling by Healthcare Providers: A Scoping Review

**DOI:** 10.3390/healthcare9050609

**Published:** 2021-05-19

**Authors:** Uchenna Benedine Okafor, Daniel Ter Goon

**Affiliations:** 1Department of Nursing Science, University of Fort Hare, 50 Church Street, East London 5021, South Africa; 2Department of Public Health, University of Fort Hare, 5 Oxford Street, East London 5021, South Africa; dgoon@ufh.ac.za

**Keywords:** prenatal physical activity, exercise, prenatal care providers, advice, counselling

## Abstract

*Background:* Despite scientific evidence on prenatal physical activity and exercise, synthesized evidence is lacking on the provision of prenatal physical activity and exercise advice and counselling by prenatal healthcare providers. The scoping review seeks to fill this gap by synthesizing available literature on the provision of prenatal physical activity and exercise advice and counselling by prenatal healthcare providers to women during antenatal visits. *Methods:* The Preferred Reporting Items for Systematic Reviews and Meta-analyses (PRISMA) search framework for scoping reviews was applied to retrieve original research articles on the prenatal physical activity and exercise practices of healthcare providers with pregnant women, published between 2010–2020, and available in English. The search databases included Google Scholar, PubMed, Science Direct, Scopus, EMBASE, The Cumulative Index for Nursing and Allied Health Literature (CINAHL), BIOMED Central, Medline and African Journal Online. Studies that fulfilled the eligibility criteria were retrieved for analysis. *Results:* Out of the 82 articles that were retrieved for review, 13 met the eligibility criteria. Seven of the articles were quantitative, four qualitative, one mixed-method and one controlled, non-randomised study, respectively. Three themes emerged as major findings. Healthcare providers affirmed their responsibility in providing prenatal physical activity advice and counselling to pregnant women; however, they seldom or rarely performed this role. Major barriers to prenatal physical activity and exercise included insufficient time, lack of knowledge and skills, inadequate or insufficient training, and lack of resources. *Conclusion*: This review highlights salient features constraining the uptake of prenatal physical activity and exercise advice/counselling by prenatal healthcare providers in both community and clinical settings. Prenatal physical activity advice and counselling are key components to the promotion of physical activity adherence during and post-partum pregnancy; this requires adequate knowledge of physical activity prescriptions and recommendations, which are personalised and contextual to environment. Research is needed to examine the prenatal physical activity advice and counselling from prenatal healthcare providers on issues hindering effective delivery of the aforementioned in the context of promoting prenatal physical activity in clinical or community settings.

## 1. Introduction

Maternal physical activity and exercise are beneficial to health and the health advantages of prenatal physical activity are widely reported in the literature. The clinical relevance of prenatal physical activity practice is incontestable. Prenatal physical activity and exercise practice lowers maternal weight gain [1,2]; decreases the risk of gestational diabetes mellitus [3,4,5,6,7], birth complications [8], fatigue, stress, anxiety and depression [9,10,11]; lowers back pain [5,7] and enhances sleep [12]. Maternal physical activity also improves breastfeeding outcomes [13]. Pregnant women can achieve these benefits by regular engagement in physical activity and exercise.

Generally, there is growing interest in physical activity counselling as a form of preventive health and treatment [14]. Scientific evidence has sufficiently proved that prenatal physical activity and exercise participation is beneficial for the mother and child [15,16,17]; therefore, the provision of physical activity and exercise advice and counselling to women, arguably, is a key component of antenatal service in maternal and clinical settings. However, one question begs examination. Do prenatal healthcare providers offer prenatal physical activity advice and counselling to pregnant women during antenatal sessions? Research studies have shown that women receive minimal or no advice or counselling from health care professionals on prenatal physical activity and exercise [18], or the advice they receive is unclear and conflicting [19]. Seemingly, there is a lack of synthesized evidence on whether the prenatal healthcare providers are knowledgeable and offer advice or counselling on prenatal physical activity and exercise to pregnant women or not. Such information and data are important as they inform strategies to improve prenatal physical activity practice. Information dissemination is a necessary strategy in changing the perception and behaviour of individuals on a particular course toward a more desirable direction or action. As gatekeepers of health concern primarily in prenatal and maternal care, prenatal healthcare providers, thus, serve as important sources of support, information and advice about prenatal physical activity and exercise [20,21]. Conversely, pregnant women regard the prenatal healthcare advice from their healthcare providers as reliable and credible [22,23,24], and they may feel motivated to make lifestyle changes; and in this case, prenatal physical activity practice for healthy pregnancy and birth, it would be beneficial. Notably, the majority of studies on prenatal physical activity counselling are based on women’s reports of prenatal care provider advice and counselling [23,25,26,27,28,29]. Therefore, the synthetization of evidence on this particular topic from the perspective of the prenatal care providers may serve to inform a best practice model on prenatal physical activity advice and counselling for prenatal healthcare providers working in primary antenatal healthcare services.

### 1.1. Scoping Review Research Question

The provision of prenatal physical activity advice and counselling is a challenge to prenatal healthcare providers. Therefore, this review seeks to answer two questions:

What are the knowledge, attitude and practice of prenatal healthcare providers regarding prenatal physical activity advice and counselling?

What barriers do prenatal healthcare providers encounter in offering prenatal physical activity advice and counselling?

### 1.2. Review Objective

The objective of this scoping review was to synthesize published studies in order to assess the knowledge, attitude and practices of prenatal healthcare providers in offering prenatal physical activity and exercise advice and counselling to women during antenatal sessions; and to further identify and characterize barriers to prenatal physical activity counselling.

## 2. Methods

### 2.1. Search Strategy

The Preferred Reporting Items for Systematic Reviews and Meta-analyses (PRISMA) Searches for scoping reviews [30] was selected as a search tool. This was a scoping review, as opposed to systematic review; therefore, it was not registered in the Prospective Register of Systematic Reviews (PROSPERO). Nonetheless, it was registered in the Open Science Framework (https://osf.io/dp8wn) (accessed on 11 May 2021). The electronic search databases included Google Scholar, PubMed, Science Direct, Scopus, EMBASE, The Cumulative Index for Nursing and Allied Health Literature (CINAHL), BIOMED Central, Medline and African Journal Online. These databases were searched to retrieve articles published from prenatal healthcare providers’ perspectives on prenatal physical activity and exercise advice and counselling published from January 2010 to December 2020. The search was updated in March 2021. The search terms and key words used were: “Prenatal physical activity”, “Prenatal exercise”, “Pregnancy”, “Pregnant women”, “Prenatal care providers”, “Healthcare providers”, “Advice”, “Counselling”, “Knowledge”, “Attitudes”, “Practices” and “Barriers”. During the data sources search process, the articles were sorted by year of publication, and then, only the first 10 pages of the search were considered because of multiple similarities (duplicates) and unrelated articles (not about prenatal physical activity counselling/not from prenatal healthcare providers). We included MeSH terms in the initial phase of the search strategy, but there was no improvement in the search and they were not utilized in the final search strategy.

### 2.2. Eligibility Criteria

As this is a scoping review, the PCC (Population/Concept/Context) framework recommended by JBI [31] was utilized to identify eligible studies. The population (P) was the prenatal healthcare providers (prenatal healthcare providers (obstetricians/gynaecologists, midwives and nurses); while the concept (C) was quantitative/qualitative studies assessing the perspectives of healthcare providers on prenatal physical activity/exercise/advice and counselling/knowledge/attitudes/practices/barriers. Pertaining to the context (C), the search process was conducted between 1 January 2010 to 31 December 2020. Only cohort, cross-section studies assessing the knowledge, attitudes, practices, and barriers of prenatal healthcare providers regarding physical activity and exercise advice and counselling across countries were included for screening and synthesis.

The exclusion criteria included non-peer-reviewed, other reviews (editorial, commentaries), book chapters, editorials, letters, and conference abstracts; abstracts without full text, articles on experiences of pregnant women regarding healthcare providers’ advice on prenatal physical activity and exercise; articles published before 2010, and those not available in English.

### 2.3. Selection Process

The PRISMA guidance on conducting and reporting scoping reviews selection process was utilized to search for articles. Scoping reviews summarise results of studies encompassing varying methodologies to identify gaps in the literature and to decipher knowledge about a phenomenon of interest [30]. Applying the eligibility criteria, two independent reviewers searched and retrieved articles in the databases. The retrieved articles were assessed for eligibility and duplication. Articles that did not meet the set criteria and were found to be duplicated were removed.

### 2.4. Data Extraction and Analysis

The titles and abstracts of the articles were screened independently by two authors (UBO and DTG) to ensure they met the eligibility criteria set for the study. The authors further performed a full text screening on the identified eligible studies from the title and abstract screening process. Based on the full text review of the selected articles, the data were further extracted in line with the objective of the study. The data were analysed applying a modified approach to thematic analysis [32]. Given the heterogeneity of the retrieved studies, a narrative, qualitative summary was provided in the form of text and tables (Table 1) to ensure comparisons among studies, assess individual study quality, rigour and theme identification [33]. Two independent authors reviewed the table and each included study. Each article was read several times and the common themes from the included studies were identified, named and recurrent themes defined. In addition, the themes were further discussed and a consensus was reached regarding the final analysis. The synthesis of the included articles’ findings that emerged from the search reflected the author(s), year of publication, setting/country, research design, sample, outcome measure(s), highlight of the main findings on prenatal healthcare providers on prenatal physical activity and exercise advice and counselling, and as well the limitations.

## 3. Results

### 3.1. Search Outcome

The search provided 82 articles. Of these, 61 articles were removed because of not meeting the eligibility criteria or being duplications. Out of the remaining 21 articles, a further eight were excluded because they were not original articles, and did not address the objective of the scoping review. Therefore, 13 articles (seven quantitative, four qualitative, one mixed methods, andone1 controlled, non-randomised study) were included in the review for analysis. Figure 1 presents a diagrammatic flow chart of the procedure for the articles’ search process and the articles screened for eligibility.

### 3.2. Characteristics of Reviewed Articles

Table 1 presents the characteristics of the 13 included studies. Most of the studies were conducted in the USA [23,34,35,36,37], three studies were conducted in the United Kingdom [38,39,40], and one each in South Africa [41], Brazil [42], Norway [43], Sweden [44], and Finland [45] respectively. Seven were quantitative studies [34,35,36,39,40,41,43], four qualitative [23,37,45], one a mixed methods study [38] and one a controlled, non-randomised study [42]. Most of the prenatal healthcare providers were midwives [37,38,39,40,43,44], obstetrician/gynaecologists [34,36,37,41] and nurses [34,37,45].

### 3.3. Themes Emerging from Studies Reviewed

#### 3.3.1. Providers’ Knowledge

Two studies reported that the providers perceived that they have a role and responsibility in providing prenatal physical activity advice and counselling [35,38,41]. Regarding the providers’ knowledge of various physical activity guidelines, two studies highlighted that providers were not familiar with the IOM [23,43] and four studies revealed the same regarding ACOG [34,39,41] guidelines. However, one study indicated provider’s familiarity with ACOG guidelines [23]. Furthermore, prenatal physical activity and exercise advice and counselling was limited and inconsistent with the then-current prenatal physical activity guidelines [23].

#### 3.3.2. Providers’ Attitudes

Regarding providers attitudes to prenatal physical activity and exercise, two studies reported that providers believed exercise during pregnancy is beneficial [34,35], whilst one study held a contrary view that vigorous exercise is not beneficial during pregnancy [34]. In addition, one study reported that providers provide individualised counselling during antenatal visits [35]. Three studies reported that providers routinely provide some counselling on physical activity and exercise during prenatal visits [23,34,35]. One study found that providers did not receive training in antenatal physical activity counselling [35]. Six studies indicated that providers advocated training on prenatal physical activity [23,34,38,40,41,46].

#### 3.3.3. Providers’ Practices

One study reported that prenatal care providers recommended exercise to their pregnant patients [34] and provided duration and intensity recommendations for patients [23]. The specific types of exercise prenatal healthcare providers recommended during pregnancy included walking and swimming [23]. One study reported that providers did describe the benefits of prenatal physical activity to women [23] and they discouraged contact sports [23].

### 3.4. Barriers to Prenatal Physical Activity Counselling

Several barriers to prenatal physical activity and exercise are synthesised in this review (Table 1). Studies cited insufficient time [23,35,38,40,44,47], lack of knowledge and skills [35,38,40] inadequate or insufficient training [23,37,38,44,45], lack of resources [35,38,40,44], lack of patient interest [23], and difficulty in counselling women with low levels of education or income or those from varying socio-cultural backgrounds [23,44] as barriers.

## 4. Discussion

This scoping review assessed the knowledge, attitudes and practices of prenatal healthcare providers in offering prenatal physical activity and exercise advice and counselling to women during antenatal sessions; and further identified and characterized the barriers to prenatal physical activity counselling. Studies on the perspectives of prenatal healthcare providers regarding physical activity counselling can inform about the unique needs of providers for appropriate interventions. Such studies are rare across countries. The review found that providers perceived that they have a role and responsibility in providing prenatal physical activity advice and counselling to pregnant women. However, the review also revealed that the providers have inadequate or no knowledge on various physical activity guidelines. This explains further why healthcare providers’ advice and counselling, in some instances, is limited and inconsistent with the then-current prenatal physical activity guidelines [23]. Prenatal healthcare providers are uniquely positioned to offer counselling and advice to pregnant woman on various health issues. They are key sources of support, information and advice about prenatal physical activity and exercise [20,21]. Women may be motivated to engage in physical activity and exercise if the healthcare providers, themselves, are knowledgeable and can offer accurate and scientific evidence-based prenatal physical activity information. This stressed the need to empower healthcare providers on prenatal physical activity education and to provide effective prenatal physical activity advice and counselling to patients; therefore, prenatal physical activity should be included in the training and curriculum of medical and health professionals concerned with maternal healthcare of women.

This review reveals prenatal care providers’ lack of knowledge about physical activity guidelines or recommendations during pregnancy as stipulated by specialised bodies and institutions such as the American College of Obstetricians and Gynaecologists (ACOG) [48], the World Health Organization (WHO) [49], the Joint Canadian Society for Exercise Physiology (CSEP)/Society of Obstetricians and Gynaecologists of Canada (SOGC) [50] and the American College of Sports Medicine (ACSM) [51]. For example, a study among midwives in the United Kingdom showed that midwives could not provide accurate information about physical activity recommendations during pregnancy, even though most midwives could confidently answer questions [39]. Previous studies elsewhere have reported similar findings, showing healthcare providers’ ignorance of physical activity/exercise guidelines [41,52,53,54]. If the prenatal care providers directly working with pregnant women were ignorant of the physical activity guideline, then, maternal physical activity would be jeopardised, as women would not be motivated to initiate, engage in and maintain physical activity and/or exercise as a long-life health activity.

The review highlights that providers believed exercise during pregnancy is beneficial to the mother and the baby [48], and to help achieve the desired outcome effectively, the providers used individualised physical activity counselling during antenatal visits [48]. Providing individualised physical activity counselling is an important step in navigating the constraining barriers to prenatal physical activity practice. Such an approach offers prenatal care providers with an opportunity to assess patients based on their specific issues affecting physical activity participation during pregnancy. Our review showed that only a few studies highlighted that prenatal care providers routinely provide some counselling on physical activity and exercise during prenatal visits [23,34,35]. Therefore, as one of the striking features of this review, the prenatal care providers advocated prenatal physical activity training [23,34,38,40,41,46]. Prenatal care providers need training on physical activity counselling and advice in order to render effective, better, evidence-based physical activity care in the antenatal practice. In addition, there is need to institutionalise prenatal physical activity as part of the mandatory antenatal care service to promote the attitude of prenatal care providers towards physical activity and exercise counselling for pregnant women.

Notably, this review highlights lack of time, and lack of knowledge and skills as the major constraining factors to prenatal physical activity counselling. A previous systematic review identified lack of time, lack of knowledge or training in physical activity counselling and lack of perceived success in changing patient behaviours as the three most cited barriers to physical activity intervention in the primary care setting [55]. In antenatal care, healthcare providers are saddled with other responsibilities, which include medical and pregnancy assessment, the conducting of antenatal tests, procedures, and bookings [56]. In some instances, they may offer counselling to women on smoking, alcohol intake and nutrition. Previous studies have shown that pregnant women who received physical activity counselling from their health providers during prenatal visits achieved higher physical activity levels compared to those who were not counselled [57,58,59]. However, this scoping review indicated that prenatal care providers rarely provide physical activity education and counselling to pregnant women during their antenatal visit. Even when the information is provided, it is inadequate or lacking in scientific content, because they have no scientific knowledge [23,34,39,41,43] nor training [23,24,37,38,45] regarding physical activity during pregnancy. This is concerning as inactive pregnant women are therefore limited information, and are likely not to be motivated to engage in physical activity. A previous study in South Africa reported that inaccurate information at the antenatal clinics impeded the ability of the health providers to provide physical activity advice to pregnant women [60]. Thus, this entails that, despite scientific evidence highlighting the benefits of prenatal physical activity, it appears that such evidence is not translatable to clinical practice. The barriers to prenatal physical activity counselling by prenatal care providers, as highlighted in this review, call for context-specific educational interventions to encourage women to practice physical activity during pregnancy.

## 5. Limitations, Strengths and Implications

The review was limited to articles published only in the English language; therefore, the possibility of other published but relevant studies in other languages having been excluded cannot be ruled out. In addition, the review focused exclusively on prenatal healthcare providers (obstetricians/gynaecologists, midwives and nurses). Therefore, the findings reported in this review cannot be extrapolated to other healthcare providers attending to women’s health. Notwithstanding these limitations, this scoping review is the first attempt to provide an overview of available evidence on prenatal physical activity counselling by prenatal healthcare providers from across multiple countries. Such studies are limited in developing countries, including Africa. The findings of this review call for concerted efforts in prenatal physical activity advice and counselling to promote the uptake of prenatal physical activity participation. This review underscores the need for prenatal care providers to prioritize the topic of physical activity and exercise during antenatal care in order to encourage pregnant women to engage in prenatal physical activity during pregnancy. Therefore, interventions to support prenatal care providers to play this key function of offering adequate counselling of physical activity pertaining to its importance, prescription and guidelines are crucial, in order to improve maternal and fetal health [50]. More studies on prenatal healthcare providers’ perspectives regarding the provision of physical activity and exercise counselling to pregnant women during pregnancy are warranted.

## 6. Conclusions

The findings of this scoping review highlight key salient points. Healthcare providers affirmed their responsibility in providing prenatal physical activity advice and counselling to pregnant women; however, they seldom or rarely performed this role. The most common barriers to prenatal physical activity counselling identified in this review are the lack of time and the requisite skill and training to advise or counsel pregnant women about prenatal physical activity during antenatal visits. Consequently, providers expressed the desire for training to improve their knowledge on prenatal activity advice or counselling. There is a need to prioritise physical activity counselling in maternal and clinical settings to encourage women to initiate and engage in physical activity during pregnancy for improved health outcomes. Prenatal care providers need time, knowledge and skill to perform this task effectively.

## Figures and Tables

**Figure 1 healthcare-09-00609-f001:**
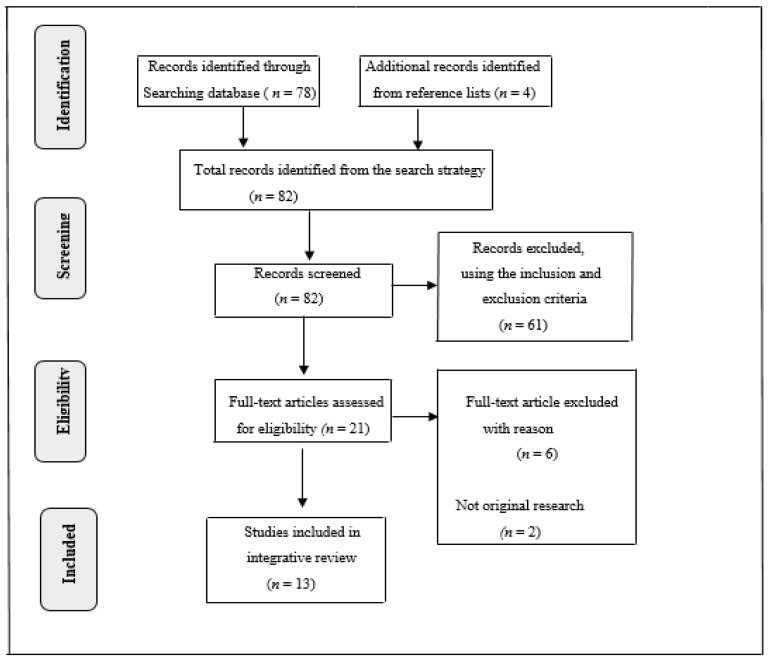
PRISMA flow diagram showing the search process for the included articles.

**Table 1 healthcare-09-00609-t001:** Summary of articles on knowledge, attitude and practices of prenatal care health providers regarding prenatal physical activity advice.

Country	Study Design	Sample and Sampling	Outcome Measure(s)	Findings	Limitations
USA [23]	Qualitative study	11 prenatal healthcare providers	Provider perceptions on physical activity counselling during prenatal care visits.	Most healthcare providers reported offering prenatal physical activity counselling during prenatal visit. Walking and swimming were the physical activities. recommended. Healthcare providers understood the benefits of prenatal physical activity counselling during pregnancy. Barriers to prenatal physical activity counselling included a lack of patient interest, lack of time, lack of training, and the low socioeconomic status of the patient. Providers were familiar with the ACOG guidelines. Advocated for further training on prenatal physical activity counselling.	Small sample size. Limited generalisability of findings. Self-reported information subject to recall and social desirability bias.
USA [34]	Cross-sectional descriptive study	Convenient sample of 93 healthcare providers.	Knowledge, beliefs, and practices of obstetric healthcare providers toward exercise during pregnancy.	Healthcare providers exhibited positive beliefs and attitudes about exercise during pregnancy. Providers recommended exercise to their pregnant patients (90%) and believed women with uncomplicated pregnancies can safely practice exercise (89%). Providers offered prenatal physical activity advice; however, the advice did not always align with the ACOG recommendations on prenatal physical activity because the providers were not familiar with the ACOG guidelines.	Non-randomisation of sample. Potential recall bias. Use of the ACOG guidelines.
USA [35]	Cross-sectional descriptive study	Convenient sample of 188 healthcare providers (91 obstetricians, 40 midwives, and 57 family medicine physicians)	Beliefs, attitudes, knowledge, self-efficacy and barriers.	Majority of the healthcare providers agreed that prenatal physical activity improves the general health of the mother and baby, and reduce excessive weight gain. They stressed the need for discussing physical activity with pregnant women; however, only about two-thirds did so with their patients. Over 40% providers were not confident with the physical activity information they had provided to women. Slightly over half of the healthcare providers offered in-office physical activity counselling, 90% accurately described the types of exercises generally considered safe for pregnant women, and 85% correctly identify the ACOG’s absolute contraindications to antenatal physical activity. Lack of time was a common barrier, and providers felt they received inadequate training.	Low response rate. Small sample from localised area. Potential recall bias.
USA [36]	Cross-sectional, retrospective study	31 Obstetricians	Association between obstetric providers’ discussions about exercise and pregnant woman exercise behaviours.	Obstetric providers’ discussion of exercise associated with patient behaviour. Obstetric providers’ ages, private insurance, number of pregnant patients seen per month; pregnant patients with complications were not associated with exercise discussion with pregnant patients.	Small sample size. Recall bias due to self-report.
USA [37]	Qualitative study	Convenience sample of 52 obstetrician/gynaecologists, midwives and nurse practitioners.	Knowledge, attitudes, and practices of prenatal care providers regarding prenatal physical activity counselling.	Providers had no training on prenatal physical activity, but relied on their own personal experiences. Prenatal physical activity was not a priority.	Small sample size. Recall bias due to self-report.
United Kingdom [38]	Multiphase mixed methods	10 randomly selected midwives	Roles, responsibilities, and barriers in providing physical activity advice and guidance to pregnant women; and any opportunities in changing pregnant women’s physical activity behaviour.	Midwives’ daily challenges affected their morale and ability to provide antenatal physical activity counselling. Midwives did not provide adequate information about prenatal physical activity. Less priority was accorded to physical activity. Barriers to prenatal physical activity advice and guidance included a lack of training, knowledge, confidence, time, and resources. Suggestions to address barriers included professional development and training, inter-professional collaboration, encouraging a support network, and challenging misconceptions about prenatal physical activity.	Small sample from localised area.
United Kingdom [39]	Descriptive online survey	59 Midwives	Midwives understanding of the NICE physical activity guidelines, and the physical activity guidance provided to women during pregnancy.	Midwives had misplaced confidence in their knowledge of the NICE physical activity guidelines for pregnancy. The positive role of exercise and benefits to mother and baby were recognised by 24% of the midwives. The majority of midwives (91%) knew about the contraindications to exercise during pregnancy, and 59% felt confident answering questions about prenatal physical activity.	Small sample size. Participants skewed to those who felt confident in their knowledge about physical activity and pregnancy. Overestimation of the findings from the midwives based on prior information about the survey.
England [40]	Cross-sectional descriptive study	192 Midwives	Barriers and facilitators associated with implementation of national guidelines for physical activity in obese pregnant woman.	Midwives perceived as not having the ability, proficiency or competency to implement, discuss and counsel women effectively on physical activity. Physical activity was not a priority. Midwives recognised their role to advise obese pregnant women about physical activity, but expressed concerns about the sensitive nature of the topic. Midwives advocated for routine prenatal physical counselling in clinical practice.	Poor response rate might result to error and bias.
South Africa [41]	Cross- sectional, descriptive study.	Convenience sample of 96 Medical Practitioners (MPs): General Practitioners (n = 58), Obstetricians/Gynaecologists (n = 33), other Specialists (n = 5).	Knowledge, attitudes and beliefs of South African MPs towards prenatal exercise.	Majority of the medical practitioners believed prenatal exercise is beneficial, but were unaware of the recommended exercise guidelines. They lacked accurate specifics about exercise prescription. Most MPs (94%) recommended moderate exercise during pregnancy. Few practitioners provided advice as well as individualised exercise prescription. Providers were not familiar with the ACOG guidelines for exercise during pregnancy. About 71% expressed a desire to attend a continuous professional development workshop on prenatal physical activity, if provided.	Low response rate. Response and selective bias. Exclusion of practitioners working in the public sector.
Brazil [42]	A controlled, non-randomized study	Doctors and nurses: Intervention group (22); Control group (20)	Effect of an educational intervention upon improving the knowledge and practices of health professionals concerning physical activity during pregnancy.	No difference in the knowledge scores between the control group and the intervention group. Compared to the control group, women in the intervention group were more likely to receive guidance regarding leisure-time walking.	Non randomisation of the sample in both groups.
Norway [43]	Cross-sectional descriptive study	65 Midwives	Midwives’ practice and views about gestational weight gain and regular physical activity and nutrition.	Physical activity advice provided at least once throughout gestation. About 32.3% midwives based their advice on personal sport/exercise experience.	Small sample size. Recall bias due to self-report.
Sweden [44]	A qualitative study	Purposive sample of 41 midwives	Swedish midwives experience about prenatal physical activity counselling; and the facilitators and barriers during pregnancy.	Barriers to prenatal physical counselling included a lack of resources and cultural expectations of the women about physical activity during pregnancy. Midwives considered the topic as sensitive for some women, especially overweight and obese women. Individualised counselling approach applied.	
Finland [45]	Descriptive qualitative approach.	Convenience sample of 11 public health nurses.	Public health nurses’ experiences of physical activity counselling.	Level of knowledge and skills about physical activity counselling was inadequate. Barrier to prenatal physical activity included women’s attitudes towards exercise, lack of time, inadequate resources, and insufficient skills. Providers suggested fmulti-professional collaboration from different healthcare areas, such as physiotherapists, physical education instructors, and dieticians. They advocated further training in physical activity.	Small sample size.

## Data Availability

Not applicable.

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
