# Peer review of "Physical Activity Advice and Counselling by Healthcare Providers: A Scoping Review"

_healthcare, 2021, doi:10.3390/healthcare9050609_

Round 1

Reviewer 1 Report

Overall, this review appears to be comprehensive and clearly explained, and addresses and important topic that will be of interest to Healthcare readers.  My comments below identify only a few minor suggestions for revision.

In the first paragraph of the introduction, there are mentions of low prenatal physical activity and this provides some motivation for the present review. It is unclear to this reviewer how that is motivating for this paper because low activity levels could just be a function of people not following medical advice about exercising.  It is impossible to know from this text whether people received exercise advice, and whether it was accurate. I would suggest the authors either clarify how this result speaks directly to the topic of provider advice/knowledge/attitudes, or just drop it as it’s confusing and raises questions that are outside the scope of the present article.

In the second part of the introduction, the key questions also seem somewhat redundant—namely are prenatal providers aware of existing physical activity guidelines and do they possess requisite knowledge seems to measure almost the same issue. Can you clarify?  The scope of the review is somewhat muddled at this point in the paper.

In terms of exclusion criteria, why were articles on the experience of pregnant women excluded? This justification could use more explanation.

Figure 1 needs to be drawn more clearly.  It is not clear why so many articles fell out of the original sample of 82?

The discussion of themes is adequate but underwhelming. It offers more of a list of results than the synthesis that was promised. Then, the following discussion seems quite long and unfocused.  The paper would be improved generally by a revision of the introduction and conclusion to provide the reader with incisive summary of the review and commentary on strengths, weaknesses, and implications.

Author Response

Reviewer #1

Comments and Suggestions for Authors

Overall, this review appears to be comprehensive and clearly explained, and addresses and important topic that will be of interest to Healthcare readers.  My comments below identify only a few minor suggestions for revision.

Response

On behalf of my co-author, I would like to express appreciation for your thorough review conducted on our manuscript. Your comments and suggestions have significantly improve the quality and outlook of the manuscript. We have tried to address the queries point by point as instructed and made appropriate amendments as necessary.

Comment

In the first paragraph of the introduction, there are mentions of low prenatal physical activity and this provides some motivation for the present review. It is unclear to this reviewer how that is motivating for this paper because low activity levels could just be a function of people not following medical advice about exercising.  It is impossible to know from this text whether people received exercise advice, and whether it was accurate. I would suggest the authors either clarify how this result speaks directly to the topic of provider advice/knowledge/attitudes, or just drop it as it’s confusing and raises questions that are outside the scope of the present article.

Response

The sentence talking to low prenatal physical activity has been deleted from the introduction as suggested. The introduction has been revised. See changes in the text in track changes.

Comment

In the second part of the introduction, the key questions also seem somewhat redundant—namely are prenatal providers aware of existing physical activity guidelines and do they possess requisite knowledge seems to measure almost the same issue. Can you clarify?  The scope of the review is somewhat muddled at this point in the paper.

Response

We agreed with your comment. The section has been revised accordingly. See changes in the text in track changes.

Comment

In terms of exclusion criteria, why were articles on the experience of pregnant women excluded? This justification could use more explanation.

Response

The articles on the experience pregnant women regarding healthcare providers’ advice on prenatal physical activity and exercise were excluded because the focus is only on the prenatal care healthcare providers’ (Population) perspectives on the provision of prenatal physical activity advice and counselling during pregnancy. 

Comment

Figure 1 needs to be drawn more clearly.  It is not clear why so many articles fell out of the original sample of 82?

Response

Figure 1 redrawn, and the reasons for articles exclusion as highlighted in the text, also included in Figure 1 appropriately.

Comment

The discussion of themes is adequate but underwhelming. It offers more of a list of results than the synthesis that was promised. Then, the following discussion seems quite long and unfocused.  The paper would be improved generally by a revision of the introduction and conclusion to provide the reader with incisive summary of the review and commentary on strengths, weaknesses, and implications.

Response

The discussion section has been revised tactfully as per your comments above. However, concerning the strengths, weaknesses, and implications, this was addressed in the paper on the section on limitations. Nevertheless, we have added further to the implications of the study thus:

This review underscores the need for prenatal care providers to prioritize the topic of physical activity and exercise during antenatal care in order to encourage pregnant women to engage in prenatal physical activity during pregnancy. Therefore, interventions to support prenatal care providers to play this key function of offering adequate counselling of physical activity pertaining to its importance, prescription and guidelines is crucial, in order to improve maternal and fetal health [50]. More studies on the prenatal healthcare providers’ perspectives regarding the provision of physical activity and exercise counselling to pregnant women during pregnancy are warranted.  

Reviewer 2 Report

GENERAL COMMENTS

Thanks to the authors for their submission to Healthcare. This manuscript aims to conduct a systematic review of the provision of antenatal physical activity and exercise counselling by antenatal healthcare providers to women during antenatal visits. I fully recognise the time and effort involved in searching and screening the data, analysing and synthesising the results, and subsequently drafting the manuscript. However, I consider that some points in the manuscript need to be attended and clarified by the authors.

COMMENTS

SUMMARY

I recommend including a sentence referring to the databases used and a brief summary of the 3 main findings.

The third paragraph of the conclusion refers to implications of the study, so I recommend omitting it from the abstract.

INTRODUCTION

I congratulate the authors for this section. It is well-written, concise and include all the needed information for contextualize using updated and relevant literature and justify the need to conduct the review.

METHODS

  • The authors declare that they follow the PRISMA guidelines in this systematic review; I recommend replacing the indicated reference with the following reference when referring to PRISMA: "Moher, D., Liberati, A., Tetzlaff, J., Altman, D. G., & Group, P. (2009). Elementos de información preferidos para revisiones sistemáticas y meta-análisis: la declaración PRISMA. PLoS Medicine, 6(7), e1000097".
  • In the section "Search strategy" the specific search strategy is not included.
  • In the Flow Diagram you refer to an additional search of the mentioned databases. If you have conducted an additional grey literature search, please indicate where this search was conducted.
  • The inclusion criteria refer to the characteristics that need to meet to be included in the review or study; while exclusion criteria refer to particular or specific characteristics that, despite meeting the inclusion criteria, could bias the review. Please, review this section and modify in consequence.
  • Was the review previously registered into PROSPERO platform? If yes, please include this information and the reference number. If not, indicate it into the method section.
  • Was the methodological quality and bias of the articles assessed? If yes, include the instrument and results please.

RESULT

3.1. Search outcome

The flowchart included into the manuscript did not follow the specific guidelines by PRISMA. I recommend the authors modifying the flowchart to PRISMA specific flow diagram.

I recommend simplifying the information included into Table 1. The table could include relevant and concise information to be visual and easy of understanding. Please, avoid complete sentence and include only key information.

The Discussion section is very long and sometimes repeats ideas. I recommend shortening its length and focusing on the 3 findings in an orderly fashion.

CONCLUSIONS

The conclusion section should be clear and concise; and should give response to the objective proposed. The inclusion of irrelevant or misplaced additional information may reduce the visibility of main conclusions and findings. Thus, sentences like "Studies on the perspectives of prenatal healthcare providers regarding physical activity counselling would inform the unique needs of providers for appropriate interventions. Such studies are rare across countries" have more sense into Discussion section. Please, consider it and modify this section in consequence.

Therefore, the manuscript has important errors that should be carefully analysed and considered by the authors before resubmission for publication.

I recommend to the authors that in future revisions they include the line number to facilitate the review process and that the tables appear in full so that they can be evaluated and contrasted with what is stated in the main text.

Kind regards,

Author Response

Reviewer #2

Comments and Suggestions for Authors

GENERAL COMMENTS

Thanks to the authors for their submission to Healthcare. This manuscript aims to conduct a systematic review of the provision of antenatal physical activity and exercise counselling by antenatal healthcare providers to women during antenatal visits. I fully recognise the time and effort involved in searching and screening the data, analysing and synthesising the results, and subsequently drafting the manuscript. However, I consider that some points in the manuscript need to be attended and clarified by the authors.

Response

On behalf of my co-author, I would like to express appreciation for your thorough review conducted on our manuscript. Your comments and suggestions have significantly improve the quality and outlook of the manuscript. We have tried to address the queries point by point as instructed and made appropriate amendments as necessary.

COMMENTS

SUMMARY

Comment

I recommend including a sentence referring to the databases used and a brief summary of the 3 main findings.

Response

We have added the databases used for the search and the included a brief summary of the three main findings as suggested.

Databases used:

The search databases included Google Scholar, PubMed, Science Direct, Scopus, EMBASE, The Cumulative Index for Nursing and Allied Health Literature (CINAHL), BIOMED Central, Medline and African Journal Online.

Brief summary:

Healthcare providers affirmed their responsibility in providing prenatal physical activity advice and counselling to pregnant women; however, they seldom or rarely performed this role. Major barriers to prenatal physical activity and exercise included insufficient time, lack of knowledge and skills, inadequate or insufficient training, and lack of resources.

Comment

The third paragraph of the conclusion refers to implications of the study, so I recommend omitting it from the abstract.

Response

Your suggested is noted; however, given the paucity of studies exploring the perspectives of healthcare providers’ on physical activity advice and counselling in the literature,  a call for more research on the subject is important in order to understand the contextual issues affecting physical activity counselling by healthcare providers during antenatal care across countries or settings.

INTRODUCTION

I congratulate the authors for this section. It is well-written, concise and include all the needed information for contextualize using updated and relevant literature and justify the need to conduct the review.

METHODS

Comment

The authors declare that they follow the PRISMA guidelines in this systematic review; I recommend replacing the indicated reference with the following reference when referring to PRISMA: "Moher, D., Liberati, A., Tetzlaff, J., Altman, D. G., & Group, P. (2009). Elementos de información preferidos para revisiones sistemáticas y meta-análisis: la declaración PRISMA. PLoS Medicine, 6(7), e1000097".

Response

The above recommended reference has been included, and the previous reference removed as suggested.

  1. Moher, D.; Liberati, A.; Tetzlaff, J.; Altman, D.G.; Group, P. Elementos de información preferidos para revisiones sistemáticas y meta-análisis: la declaración PRISMA. PLoS Med. 2009, 6, e1000097.

Comment

In the section "Search strategy" the specific search strategy is not included.

Response

We included in the section ‘Search strategy’ the search tool-PRISMA, the electronic search databases, and the search terms used to retrieve articles on prenatal counselling from January to December 2020. The search was updated in March 2021. The search terms and key words used were: ‘Prenatal physical activity’, Prenatal exercise’, ‘Pregnancy’, ‘Pregnant women’,  ‘Prenatal care providers’, ‘Healthcare providers’, ‘Advice’, ‘Counselling’, ‘Knowledge, ‘Attitudes’, ‘Practices’ and  ‘Barriers. During the data sources search process, the  articles were sorted by year of publication, and then, only considered the first 10 pages of the search because of multiple similarities (duplicates) and unrelated articles (not about prenatal physical activity counselling/not prenatal healthcare providers). We included MeSH terms in the initial phase of the search strategy, but there was no improvement in the search and were not utilized in the final search strategy.

Comment

In the Flow Diagram you refer to an additional search of the mentioned databases. If you have conducted an additional grey literature search, please indicate where this search was conducted.

Response

There was not additional grey literature search. Only peer-reviewed articles were searched and included.

Comment

The inclusion criteria refer to the characteristics that need to meet to be included in the review or study; while exclusion criteria refer to particular or specific characteristics that, despite meeting the inclusion criteria, could bias the review. Please, review this section and modify in consequence.

Response

This section has been revised as per your comment. The exclusion criteria

Hence, this is a scoping review; the PCC (Population / Concept / Context) framework recommended by JBI [31] was utilized to identify eligible studies. The population (P) was the prenatal healthcare providers (prenatal healthcare providers (obstetricians/gynaecologists, midwives and nurses); while the concept (C) was quantitative/qualitative studies assessing the perspectives of healthcare providers on prenatal physical activity/exercise/advice and counselling/knowledge/attitudes/practices/barriers. Pertaining to the context (C), the search process was conducted between 1 January 2010 to 31 December 2020. Only cohort, cross-section studies assessing the knowledge, attitudes, practices, and barriers of prenatal healthcare providers regarding physical activity and exercise advice and counselling across countries were included for screening and synthesis.

The exclusion criteria included non-peer-reviewed, other reviews (editorial, commentaries), book chapters, editorials, letters, and conference abstracts; abstracts without full text, articles on experiences of pregnant women regarding healthcare providers’ advice on prenatal physical activity and exercise; articles published before 2010, and those not available in English.

Comment

Was the review previously registered into PROSPERO platform? If yes, please include this information and the reference number. If not, indicate it into the method section.

Response

This was a scoping review, as opposed to systematic review; therefore, it was not registered in the Prospective Register of Systematic Reviews (PROSPERO).

Comment

Was the methodological quality and bias of the articles assessed? If yes, include the instrument and results please.

Response

None of the articles involves a meta-analysis or systematic review, and the outcome measure was not to assess any association; therefore, the methodological quality and bias in such manner was not performed. We only assess the articles meeting the search criteria as per the objective of the review.

RESULT

3.1. Search outcome

Comment

The flowchart included into the manuscript did not follow the specific guidelines by PRISMA. I recommend the authors modifying the flowchart to PRISMA specific flow diagram.

Response

The flow chart was designed according to PRISMA guidelines. Figure I was distorted during the submission stage─uploading of the figure. However, it has been revised.

Comment

I recommend simplifying the information included into Table 1. The table could include relevant and concise information to be visual and easy of understanding. Please, avoid complete sentence and include only key information.

Response

The information in the Table 1 has been revised for conciseness and clarity.

Comment

The Discussion section is very long and sometimes repeats ideas. I recommend shortening its length and focusing on the 3 findings in an orderly fashion.

Response

The discussion section has been shortened as suggested. See revisions in the text in tract changes.

CONCLUSIONS

Comment

The conclusion section should be clear and concise; and should give response to the objective proposed. The inclusion of irrelevant or misplaced additional information may reduce the visibility of main conclusions and findings. Thus, sentences like "Studies on the perspectives of prenatal healthcare providers regarding physical activity counselling would inform the unique needs of providers for appropriate interventions. Such studies are rare across countries" have more sense into Discussion section. Please, consider it and modify this section in consequence.

Response

The conclusion has been revised per the comment above. See revisions in the text in tract changes.

Therefore, the manuscript has important errors that should be carefully analysed and considered by the authors before resubmission for publication.

Comment

I recommend to the authors that in future revisions they include the line number to facilitate the review process and that the tables appear in full so that they can be evaluated and contrasted with what is stated in the main text.

Response

Line numbers inserted restarting on each page, and the tables revised accordingly as your suggestion.
